# Optimal Design of Acoustic Metamaterial of Multiple Parallel Hexagonal Helmholtz Resonators by Combination of Finite Element Simulation and Cuckoo Search Algorithm

**DOI:** 10.3390/ma15186450

**Published:** 2022-09-16

**Authors:** Fei Yang, Enshuai Wang, Xinmin Shen, Xiaonan Zhang, Qin Yin, Xinqing Wang, Xiaocui Yang, Cheng Shen, Wenqiang Peng

**Affiliations:** 1College of Field Engineering, Army Engineering University of PLA, Nanjing 210007, China; 2Engineering Training Center, Nanjing Vocational University of Industry Technology, Nanjing 210023, China; 3MIIT Key Laboratory of Multifunctional Lightweight Materials and Structures (MLMS), Nanjing University of Aeronautics and Astronautics, Nanjing 210016, China; 4College of Aerospace Science and Engineering, National University of Defense Technology, Changsha 410073, China

**Keywords:** broadband sound absorption, optimal design, acoustic metamaterial, hexagonal Helmholtz resonators, joint simulation method, finite element simulation, cuckoo search algorithm

## Abstract

To achieve the broadband sound absorption at low frequencies within a limited space, an optimal design of joint simulation method incorporating the finite element simulation and cuckoo search algorithm was proposed. An acoustic metamaterial of multiple parallel hexagonal Helmholtz resonators with sub-wavelength dimensions was designed and optimized in this research. First, the initial geometric parameters of the investigated acoustic metamaterials were confirmed according to the actual noise reduction requirements to reduce the optimization burden and improve the optimization efficiency. Then, the acoustic metamaterial with the various depths of the necks was optimized by the joint simulation method, which combined the finite element simulation and the cuckoo search algorithm. The experimental sample was prepared using the 3D printer according to the obtained optimal parameters. The simulation results and experimental results exhibited excellent consistency. Compared with the derived sound absorption coefficients by theoretical modeling, those achieved in the finite element simulation were closer to the experimental results, which also verified the accuracy of this optimal design method. The results proved that the optimal design method was applicable to the achievement of broadband sound absorption with different low frequency ranges, which provided a novel method for the development and application of acoustic metamaterials.

## 1. Introduction

The prevalence of low-frequency noise pollution in the modern industrial production and living has seriously affected the daily life and work of people [1]. The traditional acoustic materials, such as porous media [2] and microperforated panel [3], are commonly utilized to reduce the impact of noise, but the low frequency noise is very difficult to eliminate due to the long wavelength [4]. To block low frequency noises, the sizes of acoustic materials have to be larger than a quarter of their wavelength, but the traditional acoustic materials are hard to satisfy the special requirements of noise reduction due to the limitation of occupied space [5,6]. Thus, the novel composites with special physical characteristics have become the research focus in the fields of noise reduction and sound absorption [7]. For example, the acoustic metamaterial with subwavelength physical properties that traditional acoustic materials do not possess has attracted great attention among various scholars, which can provide novel methods to reduce the low frequency noise effectively.

In the latest years, the Helmholtz resonators with extended aperture depth have been a favorable choice for the low frequency noise absorption [8,9,10,11,12]. Jimenez et al. [8] proposed realistic panels which were made of Helmholtz resonator bricks, and the perfect absorption of sound at the frequency of 338.5 Hz was achieved. The acoustic metamaterial absorber composed of thick-necked embedded Helmholtz resonators was developed by Zhang and Xin [9] to achieve the perfect sound absorption at the 150 Hz. Sharafkhani [10] presented a multi-band sound absorber based on the Helmholtz resonators, and it could achieve the perfect absorptions at 100 Hz, 200 Hz, and 300 Hz simultaneously. However, the relatively narrow absorption bandwidth of a single Helmholtz resonator cannot satisfy practical engineering applications. Moreover, the material thickness is usually limited to small installation spaces. To overcome these problems, the prominent strategy involved in the embedded multiple individual single Helmholtz resonators with various sizes were proposed in this research. The overall bandwidth of the entire acoustic material is expanded by overlapping the absorption bandwidth of each single Helmholtz resonator.

For the practical applications, the design of acoustic materials with the desired sound absorption effect is the most important goal. Thus, substantial research has been carried out on optimizing the geometric parameters of the acoustical materials [13,14,15,16,17]. Gao et al. [13] proposed a hybrid design of unit cells consisting of multiple labyrinthine channels, and a genetic algorithm was used to optimize the absorption performance so that it could have best absorption energy in the frequency range of interest. Yan et al. [14] adopted the particle swarm algorithm to optimize the depth of the honeycomb core cavity of the honeycomb micro-perforated plate (HMPP). The optimization results showed that the HMPP structure had the multiple peak absorption coefficients in the range of 0–3500 Hz, and the absorption coefficient was above 0.85 in a wide frequency band. Gao et al. [15] had optimized the ultra-broadband parallel sound absorber (UBPSA) using a teaching–learning-based optimization algorithm, in which geometric parameters were utilized as the optimization variables. The optimized UBPSA showed the high absorption effect in the frequency range of 200–1715 Hz. Currently, most studies on the optimization of acoustic materials have adopted the acoustic–electric analogy or transfer matrix method for theoretical modelling, followed by the combination of the intelligent algorithms for optimization, which have greatly reduced the optimization time. However, these methods have not been able to accurately optimize the acoustic metamaterial. Due to the immature research on the acoustic absorption mechanism of acoustic metamaterial [13,14,15,16,17], a number of simplifications are usually taken into the process of the theoretical modeling, resulting in large errors between the theoretical results and experimental results. Hence, the accuracy of the optimization results may not be precise. To overcome these shortcomings, the joint simulation method incorporating the finite element simulation and cuckoo search algorithm was adopted in this study, which had attempted to improve the accuracy of the optimization result.

In this study, the design method of acoustic metamaterial of multiple parallel hexagonal Helmholtz resonators with sub-wavelength dimensions was discussed. The optimized acoustic metamaterials could achieve the ideal broadband sound absorption effect at a low frequency range in a small space. First, in order to ease the optimization burden and speed up the optimization process, the overall thickness, aperture size, and side length of the cavity for the acoustic metamaterial were determined depending on the application scene and processing difficulty. The depth of neck was initially selected by the two-dimensional finite element analysis. Then, a joint simulation method incorporating finite element analysis and cuckoo search algorithm was conducted to obtain the optimal geometric parameters, which satisfied the sound absorption requirement. The optimization took the depth of neck as the optimization parameter and the total absorbed sound energy at the target frequency range as the optimization target. Finally, samples were prepared to obtain experimental data according to the optimized parameters. The optimized parameters were also substituted into a theoretical model to obtain the theoretical data. By comparing the theoretical data, simulation data, and experimental data, the accuracy and reliability of the proposed optimization method were verified. In addition, the applicability of this optimal design method for the different conditions was also further discussed, which provided a novel method for the development of similar acoustic metamaterial.

## 2. Initial Structural Design

### 2.1. Design Objective

A noise test was conducted on a workshop under the operation mode. The test results indicated that major frequencies of the noise were concentrated in the range of 487–675 Hz. The acoustic materials would exhibit a better sound absorption performance in the simulation environments than in the reality of applications due to the fabrication errors and actual installation conditions. Thus, the simulation target was set as slightly larger in the design stage. To realize all the actual sound absorption coefficients larger than 0.8 in the interested frequency range, sound absorption coefficients of the designed acoustic metamaterial were required to be above 0.85 in the frequency range of 420–700 Hz during the simulation process.

### 2.2. Geometric Parameters Design

In order to reduce the noise and improve the indoor acoustic environment in the given workshop, the acoustic metamaterial of multiple parallel hexagonal Helmholtz resonators was proposed in this research, as shown in Figure 1.

The schematic diagram of the whole structure for the designed acoustic metamaterial is shown in Figure 1a. The acoustic metamaterial consisted of an array of nineteen hexagonal Helmholtz resonators, each of which was a hexagonal cavity structure that contained a microporous plate embedded in an extended neck and rear cavity, as shown in Figure 1b. The nineteen single hexagonal Helmholtz resonators were divided into five groups to reduce the computational burden, with structures 1–3 as the first group, structures 4–7 as the second group, structures 8–12 as the third group, structures 13–16 as the fourth group, and structures 17–19 as the fifth group, as shown in Figure 1a. Figure 1c,d showed details of a single hexagonal Helmholtz resonator, where a is the side length of a single hexagonal resonator; t is the thickness of wall; $d_{i}$ is the diameter of the micropore; $l_{i}$ is the depth of the neck; L is the total thickness of the acoustic metamaterial.

The various restrictions would be placed on the data of design variables to meet the certain usage requirements in the subsequent optimization process. Excessive variables would lead to an exponential increase in the calculation effort for the simulation and the optimization, so a certain size design should need to be predetermined.

To ensure the low frequency sound absorption, the perforation rate should not be too high. Meanwhile, taking the widest possible sound absorption band into consideration, the perforation rate should not be too low [18,19,20]. Therefore, the perforation rate of the five groups was selected to be in the range of 0.5–0.9% [20], and the size of the perforation was ensured to be kept different among the various groups. Therefore, for the five groups of hexagonal Helmholtz resonator in Figure 1a, the derived diameters of micropore were 4.47 mm, 3.54 mm, 3.75 mm, 4.74 mm, and 5.16 mm, respectively, which corresponded to the perforation rate of 0.6%, 0.5%, 0.7%, 0.9%, and 0.8% successively. Because the designed acoustic metamaterial would be installed between the building decorative panels and the roof, the thickness of acoustic metamaterial should not be too large, and here the total thickness L was limited to be 40 mm. To reduce the overall structure mass, the thickness of wall t was taken to be 2 mm. The side length a of the single Helmholtz resonator was set to be 10 mm by taking into account the requirement of standing wave tube test for actual sound absorption coefficients of the investigated acoustic metamaterial.

With these three already determined structural parameters, the ideal sound absorption effect was achieved by adjusting the depth of neck for each single structure. Without taking the mutual coupling effect among the various Helmholtz resonators into account, each single structure corresponded to generate one resonant frequency. The depth of neck was adjusted to allow nineteen resonant frequencies at around 420–700 Hz. The resonant frequencies were 425 Hz, 440 Hz, 455 Hz, 470 Hz, 485 Hz, 500 Hz, 515 Hz, 530 Hz, 545 Hz, 560 Hz, 575 Hz, 590 Hz, 605 Hz, 620 Hz, 635 Hz, 650 Hz, 665 Hz, 680 Hz, and 695 Hz as an interval of 15 Hz was taken for the division, which were aimed to be obtained by hexagonal Helmholtz resonator of 4, 5, 6, 7, 8, 9, 10, 11, 12, 1, 2, 3, 13, 14, 15, 16, 17, 18, and 19, respectively, in Figure 1a.

To visually illustrate the design process of initial value of the depth of neck, structure 4, corresponding to the resonant frequency 425 Hz, was taken as an example. First, a cylindrical representative hexagonal structure with the approximate boundary conditions was chosen to achieve the desired sound absorption properties of acoustic materials [21,22]. The radius of cylinder is equal to $\sqrt{S/\pi }$, where S is the cross-sectional area of the hexagon in Figure 1c and is calculated as $\frac{3\sqrt{3}}{2}a^{2}$. In order to decrease the computational time and improve the research efficiency, the rotational 3D model could be further simplified to a 2D model for the finite element analysis, as shown in Figure 2. Afterwards, through setting the parametric scan of the depth of neck in the range of 10–15 mm with the interval of 0.5 mm to obtain the absorption curve in the frequency range of 420–430 Hz, an approximate range of value was obtained as [14.5 mm, 15 mm]. To further confirm the value of the depth of neck, the parameter scan of the depth of neck was performed in the approximate range of value [14.5 mm, 15 mm] with the interval of 0.1 mm. Finally, the initial depth of neck of structure 4 was determined to be 14.7 mm, which could obtain a resonant frequency 425 Hz in theory. By analogy, the initial geometric parameters of overall structure were obtained and summarized in Table 1.

According to the initial geometric parameters in Table 1, the finite element simulation model of the designed acoustic metamaterial of multiple parallel hexagonal Helmholtz resonators was obtained, and distributions of its sound absorption coefficients are shown in Figure 3. Judging from Figure 3, the sound absorption coefficients of the initial structure were larger than 0.85 in the frequency range of 496–686 Hz, which was far from the desired target frequency band 420–700 Hz. Therefore, the further optimization of geometric parameters for the designed acoustic metamaterial of multiple parallel hexagonal Helmholtz resonators was essentially required to satisfy the noise reduction for the given conditions in this research.

## 3. Optimization of Acoustic Metamaterial

### 3.1. Establishment of Finite Element Simulation Model

Finite element simulation was a prerequisite for the optimization of the acoustic metamaterial by using optimization algorithms. In this study, the numerical model of acoustic metamaterial was constructed by COMSOL [23,24,25]. The constructed finite element model based on the thermo-viscous acoustic module is shown in Figure 4.

The components of the numerical model were identified in the figure. The background pressure field was simulated as a plane wave sound field with the sound pressure value of 1 Pa and the sound velocity of 343 m/s. The input sound waves were incident vertically on the surface of the acoustic metamaterial with the negative direction of the *z*-axis. The perfect match layer was added at the end of the background pressure field for full absorption of the reflected acoustic waves, which prevented the acoustic reflection from affecting the calculation results. The thickness of the perfect match layer was 1.5 times of that of background pressure field, and their sizes were determined by total thickness of the acoustic absorber and the required frequency range simultaneously. Generally speaking, their sizes would be larger when the total thickness of the acoustic absorber was larger or the analyzed frequency range was in a lower region. In this study, thickness of the background was 8 mm, and that of the perfect match layer was 12 mm. The model material was set as air, and the material properties used the default values in the COMSOL.

Moreover, in the numerical model, the mesh of the perfect match layer was divided into 6 layers through sweeping, and the rest were tetrahedral meshes, as shown in Figure 4b. Each wavelength must be divided into at least six meshes to ensure computational accuracy. Therefore, the utilized parameters of finite element simulation model in this research were as follows: size of the biggest unit, 3.5 mm; size of the smallest unit, 0.15 mm; the maximum growth rate of neighboring unit, 1.35; curvature factor, 0.3; resolution ratio of the narrow area, 0.85; mesh type, free tetrahedron mesh; layer number of the boundary area, 5; stretching factor of the boundary layer, 1.2; regulatory factor of the thickness of the boundary layer, 1. Furthermore, the frequency domain solver was chosen for the calculation, and the investigated frequency range was selected as 200–1600 Hz.

### 3.2. Optimization Algorithm

The cuckoo search algorithm [26,27,28,29,30,31,32] was chosen as the optimization algorithm in this research, which performed a global search by simulating the parasitic brood behavior of the cuckoo nests using Lévy flight. The algorithm had an excellent global optimization seeking capability. Currently, various studies and applications have demonstrated that the cuckoo search algorithm is very effective in solving scheduling problems and combinatorial optimization problems. Therefore, it was quite suitable for solving the optimization problems of acoustic metamaterial under the limited conditions.

The acoustic metamaterial was optimized by the cuckoo search algorithm in order to achieve an absorption coefficient in the target frequency range 420–700 Hz greater than 0.85. Therefore, the optimization process was divided into the following five steps.

(1)Set the initial parameters for the algorithm. In the current optimization process, the number of host nest populations was set as N=20, the maximum number of iterations was N_iterTotal=100, and the maximum probability of discovery was $p_{a}=0.25$.(2)Calculate the fitness function of the population individual. In this research, sound absorption performance within the target frequency range of the acoustic metamaterial of multiple parallel hexagonal Helmholtz resonators was the research object and the depth of neck of each resonator was the optimization parameter. To ensure that the resulted absorption curve could maintain a high absorption coefficient in the target frequency range, the fitness value function max(α) was chosen as the total amount of sound energy absorbed at the target frequency range, as shown in Equation (1).


(1)
$$\max(\alpha )=\int_{f_{0}}^{f_{1}}\alpha (f)df$$


The corresponding discrete form of the fitness function was as follows.
(2)$$\max(\alpha )=\sum_{i=1}^{n}\alpha_{i}\Delta f$$

In Equations (1) and (2), α(f) is the sound absorption coefficient for the corresponding frequency f; Δf is the given adjacent frequency interval; $f_{0}$ and $f_{1}$ are the upper and lower limit of the target frequency band, respectively; n is the number of frequency intervals. 

(3)Update all nests according to Equation (3).


(3)
$$x_{i}^{t+1}=x_{i}^{t}+\partial ⊕L(\lambda )$$


Here, $x_{i}^{t+1}$ is the ith bird nest at the tth iteration; ∂ denotes step control amount, and $L(\lambda )=t^{-\lambda }(1<\lambda \leq 3)$ is the Lévy random search path. Then, better individuals were selected by assessing fitness.

(4)For all nests $x_{i}(i=1,\cdots ,N)$, a random number $r_{i}\in [0,1]$ is generated and compared to the probability $p_{a}$. If $r_{i}>p_{a}$, then $x_{i}^{t+1}$ is randomly changed, otherwise there will be no change. The adaptation of the nests before and after the random change was evaluated, retaining the nest with better adaptation as the final $x_{i}^{t+1}$. Afterwards, the return step (2) was iterated.(5)When all the absorption coefficients at the target frequency range were above 0.85, or it had reached the maximum number of iterations, the iteration ended, and the current optimal individual was output.

The optimization process of the cuckoo search algorithm for the acoustic metamaterial of multiple parallel hexagonal Helmholtz resonators is shown in Figure 5. 

### 3.3. Interactive Operations with MATLAB and COMSOL

COMSOL had a powerful multi-physics field simulation capability and weak ability to optimize physical field parameters by using the intelligent algorithms. However, COMSOL’s powerful interface compatibility could construct a joint simulation platform with MATLAB to realize the joint simulation of algorithms and physical fields.

First, the acoustic finite element simulation model was constructed in COMSOL and the other physical parameters, except these optimization parameters were set. The project was exported to an m-file by using the ‘Save As’ option. Next, MATLAB was opened by “COMSOL with MATLAB” and it operated the COMSOL automatically in the background. The cuckoo search algorithm was combined with the m-file of the acoustic finite element simulation model in MATLAB to realize the control of the algorithm on the physical field simulation. The process of using the joint simulation platform for algorithm and physical field simulation was summarized and is shown in Figure 6.

## 4. Results and Discussion

### 4.1. Optimization Results

The geometric parameters for the optimized acoustic metamaterial of multiple parallel hexagonal Helmholtz resonators are summarized and shown in Table 2. The optimal sound absorption coefficients with their corresponding nineteen single Helmholtz resonator structures are shown in Figure 7a. It could be judged from Figure 7a that the sound absorption curve met the design requirement of a sound absorption coefficient above 0.85 in the frequency range of 418–709 Hz. The absorption peaks were 0.92917, 0.89134, 0.95469, 0.95203, 0.89158, 0.88892, 0.89092, 0.93914, and 0.91841 corresponding to resonant frequency approximately 432 Hz, 463 Hz, 538 Hz, 545 Hz, 571 Hz, 600 Hz, 633 Hz, 660 Hz, and 694 Hz, respectively. The overall thickness of the acoustic material was only 40 mm, which was near 1/20 of the wavelength of the first peak absorption (*λ* = *c*/*f* = 343/432 = 0.794 m), meaning that the acoustic metamaterial had the ability to absorb the sound in the sub-wavelength range. The acoustic metamaterial had better sound absorption performance after the optimization. The absorption peaks of the combined structure were larger than the absorption peaks of single Helmholtz resonators, which indicated that the absorption curve of the combined structure was not a simple superposition of the resonant frequencies of the nineteen single structures, but a common coupling effect of the nineteen structures. The coupling effect between the various structures improved the sound absorption effect of the combination structure and achieved a broadband efficient sound absorption. Because the peak frequencies of the absorption coefficients corresponding to the different depths of neck were close to each other, the curve of the sound absorption coefficient for this acoustic metamaterial of multiple parallel hexagonal Helmholtz resonators was smooth.

In order to reveal the sound absorption mechanism of acoustic metamaterial clearly, the instantaneous local velocity of the corresponding absorption peaks was presented in Figure 7b. As shown in Figure 7b, the highlighted areas were concentrated on the necks. This indicated that the velocity of air particles in the necks was much higher than that of air particles in the rear cavities due to the geometric discontinuity. The thermal viscous effect between the air and the walls of necks converted the acoustic energy into thermal movement and achieved noise attenuation. It could be observed more visually that the acoustic metamaterial achieved the broadband sound absorption effect at the low frequency through the common coupling effect of multiple hexagonal Helmholtz resonators. The closer to the low frequencies, the more hexagonal Helmholtz resonators were involved in sound absorption. Because of the insufficient sound absorption capacity of a single hexagonal Helmholtz resonator within the low frequency range, multiple hexagonal Helmholtz resonators were required to work together to achieve the desired sound absorption effect.

### 4.2. Theoretical Analysis

For further verification of the accuracy and reliability of the optimal design, theoretical modeling of the acoustic metamaterial was also carried out [33,34,35]. The acoustic impedance Z of the overall structure of acoustic metamaterial of multiple parallel hexagonal Helmholtz resonators with sub-wavelength dimension consists of the acoustic impedance $Z_{i}$ of all nineteen single Helmholtz resonators connected in parallel, which can be obtained by Equation (4). The acoustic impedance $Z_{i}$ of single Helmholtz resonator consists of the acoustic impedance of the perforated plate $Z_{im}$ and that of cavity $Z_{ic}$, as shown in Equation (5). The perforated plate impedance $Z_{im}$ can be calculated by Euler’s equation, as shown in Equation (6).
(4)$$Z=\frac{1}{\sum_{i=1}^{19}\left(1/Z_{i}\right)}$$
(5)$$Z_{i}=Z_{im}+Z_{ic}$$
(6)$$Z_{nm}=i\frac{\omega \rho_{0}l_{i}}{\sigma_{i}}{\left[1-\frac{B_{1}\left(\frac{d_{i}}{2}\sqrt{-i\frac{\rho_{0}\omega }{\mu }}\right)}{\left(d_{i}\sqrt{-i\frac{\rho_{0}\omega }{\mu }}\right)\cdot B_{0}\left(\frac{d_{i}}{2}\sqrt{-i\frac{\rho_{0}\omega }{\mu }}\right)}\right]}^{-1}+\frac{\sqrt{2\mu \rho_{0}\omega }}{2\sigma_{i}}+i\frac{0.85\omega \rho_{0}\cdot d_{i}}{\sigma_{i}}$$
where $\sigma_{i}$ is the perforation rate corresponding to a single Helmholtz resonators cavity; $l_{i}$ is the perforation length; $\rho_{0}$ is the density of air at room temperature and pressure; $B_{1}\left(\frac{d_{i}}{2}\sqrt{-i\frac{\rho_{0}\omega }{\mu }}\right)$ and $B_{0}\left(\frac{d_{i}}{2}\sqrt{-i\frac{\rho_{0}\omega }{\mu }}\right)$ are the first-order and zero-order first-class Bessel functions, respectively; μ is the dynamic viscosity coefficient of air; ω is the acoustic angular frequency and f is the acoustic frequency. The cavity impedance $Z_{ic}$ is gained by the impedance transfer equation, as shown in Equation (7).
(7)$$Z_{ic}=-i\rho_{ce}c_{ce}A/A_{c}\cot\left(k_{ce}L\right)$$
where A is the area of the whole structure; $A_{c}$ is the cross-sectional area of the cavity, $\rho_{ce}$ is the effective density of air, which can be obtained from Equation (8); $c_{ce}$ is the effective volume compression coefficient of air, which can be gained from Equation (9); $k_{ce}$ is the effective transfer constant of air in the cavity, which can be obtained from Equation (10); L is the length of the rear cavity.
(8)$$\rho_{ce}=\rho_{0}{\left(1-\frac{2B_{1}\left(r_{i}\sqrt{-i\frac{\rho_{0}\omega }{\mu }}\right)}{\sqrt{-i\frac{\rho_{0}\omega }{\mu }}B_{0}\left(r_{i}\sqrt{-i\frac{\rho_{0}\omega }{\mu }}\right)}\right)}^{-1}$$
(9)$$c_{ce}=\frac{1}{\gamma P_{0}}\left(\gamma -\left(\gamma -1\right)\frac{1-2{\left(\sqrt{-i\omega \frac{\rho_{0}C_{p}}{\kappa }}\right)}^{-1}B_{1}\left(r_{i}\sqrt{-i\omega \frac{\rho_{0}C_{p}}{\kappa }}\right)}{B_{0}\left(r_{i}\sqrt{-i\omega \frac{\rho_{0}C_{p}}{\kappa }}\right)}\right)$$
(10)$$k_{ce}=\omega \sqrt{\rho_{ce}c_{ce}}$$
where $C_{p}$ is the specific heat at the constant pressure; γ is specific heat ratio, $P_{0}$ is standard atmospheric pressure at room temperature and pressure.

The total sound absorption coefficient could be determined by Equation (11).
(11)$$\alpha =1-{\left|\frac{Z-\rho_{0}c_{0}}{Z+\rho_{0}c_{0}}\right|}^{2}$$
where $c_{0}$ is the sound speed of air at room temperature and pressure, 343 m/s.

### 4.3. Experimental Validation

#### 4.3.1. Methodology

To verify the sound absorption effect of the acoustic metamaterial, experiments were conducted with the AWA6290T transfer function sound absorption coefficient measurement system (supported by Hangzhou Aihua Instruments Co., Ltd., Hangzhou, China), as shown in Figure 8a, which could detect the sound absorption coefficients of sound absorbing material or structures with normal incidence according to GB/T 18696.2-2002 (ISO 10534-2:1998) “Acoustics—Determination of sound absorption coefficient and impedance in impedance tubes—part 2: Transfer function method”, and its schematic diagram is shown in Figure 8b [36,37,38,39,40,41]. A cylindrical sample with a diameter of 100 mm was manufactured by the low force stereolithography (LFS) 3D printer of Form3 (supported by the Formlabs Inc., Summerville, MA, USA), as shown in Figure 8c, and the prepared sample for the investigated acoustic metamaterial is exhibited in Figure 8d. The proposed acoustic metamaterial of the multiple parallel hexagonal Helmholtz resonators was modeled in the 3D modeling software, and it was further introduced into the Preform software supported by Form3 3D printer. When fabrication of the sample was finished, it was further cleaned by the Formlabsform wash (Formlabs Inc., Boston, MA, USA) to remove residual liquid resin and irradiated for solidification by the FormlabsForm Cure (Formlabs Inc., Boston, MA, USA). The used photosensitive liquid resin in this research was ClearV4, which was purchased from the self-support flagship store of Formlabs 3D printer in JD.com (JD.com Inc., Beijing, China). The acoustic metamaterial made by photosensitive resin 3D printing had a smooth surface and well hardness, which met experimental requirements [36,37,38,39,40,41]. The AWA6290T detector consisted of the AWA5871 power amplifier, the AWA6290B dynamic signal analyzer, the AWA8551 impedance tube, and corresponding analysis software in the workstation, as shown in Figure 8b. The analysis software could finish the 1/3 OCT analysis and fast Fourier transform (FFT) analysis. Meanwhile, the original incident acoustic wave was also controlled by the signal generation software in the workstation. The detected sample was fixed in the end of the impedance tube, and two microphones were utilized to detect the sound pressure of the incident and reflected acoustic waves, which could derive the sound absorption coefficient at certain frequency according to the transfer function method [36,37,38,39,40,41]. The distance between the two microphones was set as 70 mm. The detected frequency range was 200–1600 Hz and there were 1502 sampling frequency points in this range. Moreover, for the purpose of elimination of the accidental error, the detection was repeated for 200 times for each sampling frequency point, and the final data were an average of the 200 values obtained in the 200 times of measurement. The measurement process was fully automatic, and took no more than 10 min for one detection [20].

#### 4.3.2. Experimental Results

Comparisons of the theoretical, simulation and experimental sound absorption coefficients of the proposed acoustic metamaterial of multiple parallel hexagonal Helmholtz resonators are shown in Figure 9. From the curve of experimental data, it was clear that the wave peaks of the absorption coefficients measured experimentally for the acoustic metamaterial were 0.92117, 0.94360, 0.8867, 0.87560, 0.86980, 0.94540, and 0.92770, respectively, which occurred at a frequency of approximately 435.059 Hz, 533.203 Hz, 565.43 Hz, 593.262 Hz, 621.094 Hz, 654.053 Hz, and 687.012 Hz successively. The deviations between the simulation data and experimental data might be due to the imperfection of the sample fabrication process, the inevitable gap between the test sample, and the inner wall of the impedance tube. However, the general view of the two curve trends for simulation data and experimental data exhibited better consistency relative to the theoretical data. This research defined the absorption coefficient above 0.8 as the effective absorption coefficient. Both experimental and simulation data showed that the acoustic metamaterial could achieve effective absorption regarding the noise frequency in the workshop, which verified the accuracy of the optimal design method.

The theoretical data of the optimized structure were obtained according to theoretical model in Section 4.2, as shown in Figure 9. It could be seen from Figure 9 that the deviation of the theoretical data from the experimental data was larger compared with the simulation data. The deviation of the theoretical data and simulation data relative to the experimental data at the target frequency range was quantitatively evaluated by the mean absolute deviation, as shown in Equations (12) and (13), respectively. The deviation of the theoretical data was calculated to be 0.0682, while that of the simulation data was 0.0203. The results showed that the simulation data were better than the theoretical data. This might be because in theoretical calculations, the cavities of the acoustic metamaterial were connected in parallel, and the impedances of each cavity were not affected by each other. However, during the simulation and experiment process, the cavities of the acoustic metamaterial were separated by a certain thickness of wall and affected each other inevitably [12]. Thus, the impedance of the cavities in different regions was different. This also further verified the accuracy and reliability of the optimal design method.
(12)$$Dev1=average\left(\left|\alpha_{simulation}\left(f\right)-\alpha_{\exp erimental}\left(f\right)\right|\right)\;f\in \left[f_{\min},f_{\max}\right]$$
(13)$$Dev2=average\left(\left|\alpha_{theoretical}\left(f\right)-\alpha_{\exp erimental}\left(f\right)\right|\right)\;f\in \left[f_{\min},f_{\max}\right]$$

### 4.4. Applicability of the Optimal Design

The accuracy and reliability of the proposed optimal design method were previously verified. Meanwhile, the applicability of the optimal design method was further explored in this research. Two additional application conditions with different noise reduction requirements were selected and samples were prepared for experimental testing based on the optimized parameters. The design requirement of Condition-1 was to attenuate noise at 900–1300 Hz within a space of only 20 mm. The design requirement of Condition-2 was to attenuate the noise at 600–1000 Hz within a space of 30 mm. 

The optimized geometric parameters for the design requirement of Condition-1 are shown in Table 3. The corresponding experimental data are shown in Figure 10, which showed that the noise in the range of 842–1355 Hz was attenuated with the sound absorption coefficient above 0.8 in accordance with the requirements of Condition-1.

The geometric parameters optimized for the design requirement of Condition-2 are shown in Table 4. The corresponding experimental data are shown in Figure 11, which showed that the noise in the frequency range 599–1027 Hz was attenuated with the sound absorption coefficient above 0.8, in accordance with the requirements of Condition-2.

It demonstrated that optimal design was still effective when the application condition changed. The ability to adjust the sound absorption band of the structure proved the applicability of this proposed optimal design method. It provided a novel method for the design of acoustic metamaterials, with potential application prospects in the fields of noise reduction and sound absorption.

## 5. Conclusions

In summary, an optimal design method to develop the acoustic metamaterials with broadband sound absorption at the low frequency range was proposed. An acoustic metamaterial of multiple parallel hexagonal Helmholtz resonators with the sub-wavelength dimension was employed. Initial parameters of the structure were designed preliminarily, and they were further optimized by the joint simulation method incorporating finite element analysis and cuckoo search algorithm. Subsequently, the theoretical results, simulation results, and experimental results were investigated, respectively.

(1)For the problem of low frequency broadband noise in the workshop and the narrow absorption bandwidth of a single Helmholtz resonator, the acoustic metamaterial of multiple parallel hexagonal Helmholtz resonators was designed and optimized. Its effective absorption capacity was verified by experimental validation. Therefore, the proposed acoustic metamaterial could attenuate the noise with a broad frequency range, which was also beneficial in practical engineering applications.(2)A joint simulation method incorporating the finite element simulation and cuckoo search algorithm was employed for optimization of the proposed acoustic metamaterial to achieve broadband sound absorption effect at the low frequency range in this research. Compared with the optimization method based on theoretical models, the design method proposed in this research could obtain more accurate optimization results, which satisfied the requirements of certain practical conditions.(3)The optimal design method proposed in this research could be applied to the absorption needs for different conditions, which also proved its feasibility and practicality. It presented a novel method for the development and application of acoustic metamaterial, which would be favorable to promote its practical applications.

## Figures and Tables

**Figure 1 materials-15-06450-f001:**
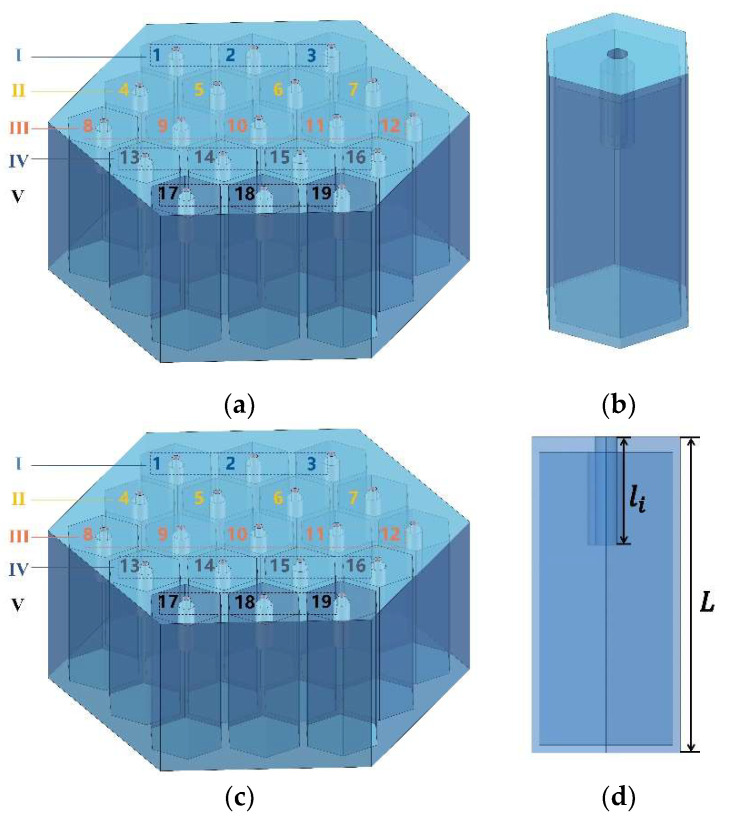
The designed acoustic metamaterial of multiple parallel hexagonal Helmholtz resonators. (**a**) Schematic diagram of the whole structure; (**b**) Structure of a single hexagonal Helmholtz resonator; (**c**) Top view of the single hexagonal Helmholtz resonator; (**d**) Main view of the single hexagonal Helmholtz resonator.

**Figure 2 materials-15-06450-f002:**
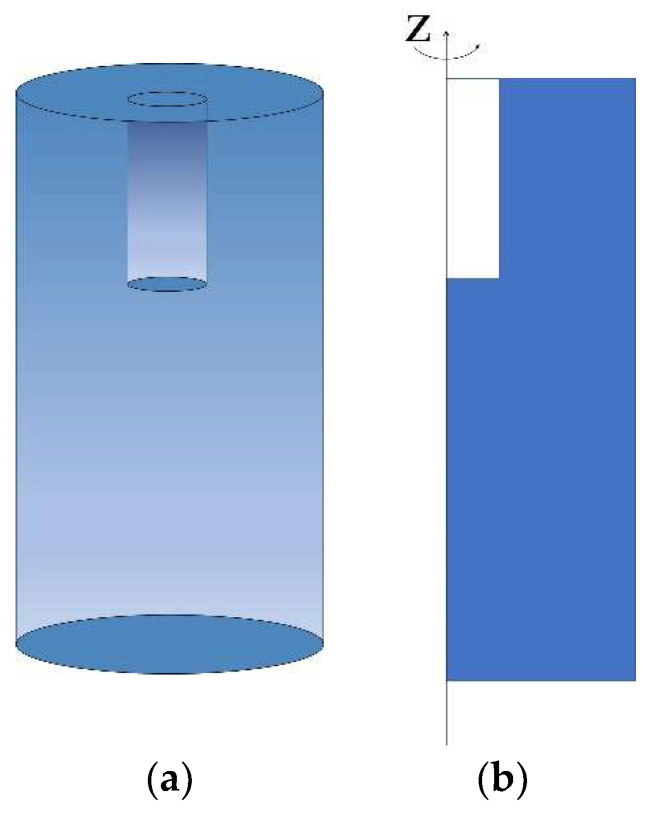
The single Helmholtz resonator. (**a**) Schematic diagram of the cylindrical representative hexagonal structure; (**b**) Schematic diagram of the 2D rotationally symmetric model for the finite element analysis.

**Figure 3 materials-15-06450-f003:**
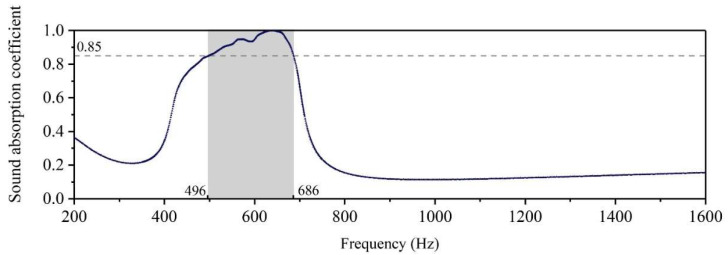
Sound absorption coefficient of the initial structure gained by finite element simulation. The gray area represents the frequency range that meets the design requirements.

**Figure 4 materials-15-06450-f004:**
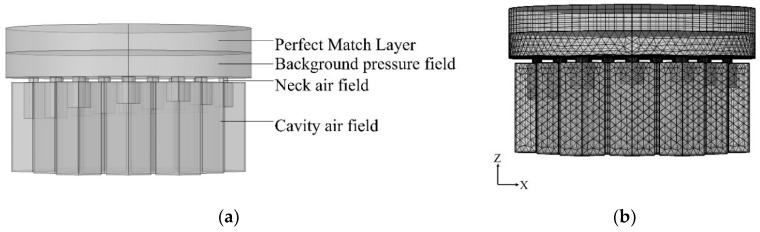
Finite element simulation model. (**a**) Finite element model of the acoustic metamaterial of multiple parallel hexagonal Helmholtz resonators. (**b**) Finite element mesh division of the proposed acoustic metamaterial.

**Figure 5 materials-15-06450-f005:**
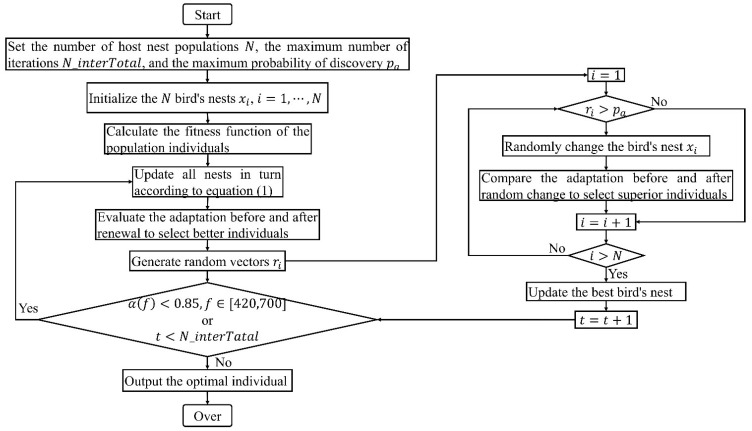
The optimization process of cuckoo search algorithm for the acoustic metamaterial of multiple parallel hexagonal Helmholtz resonators.

**Figure 6 materials-15-06450-f006:**
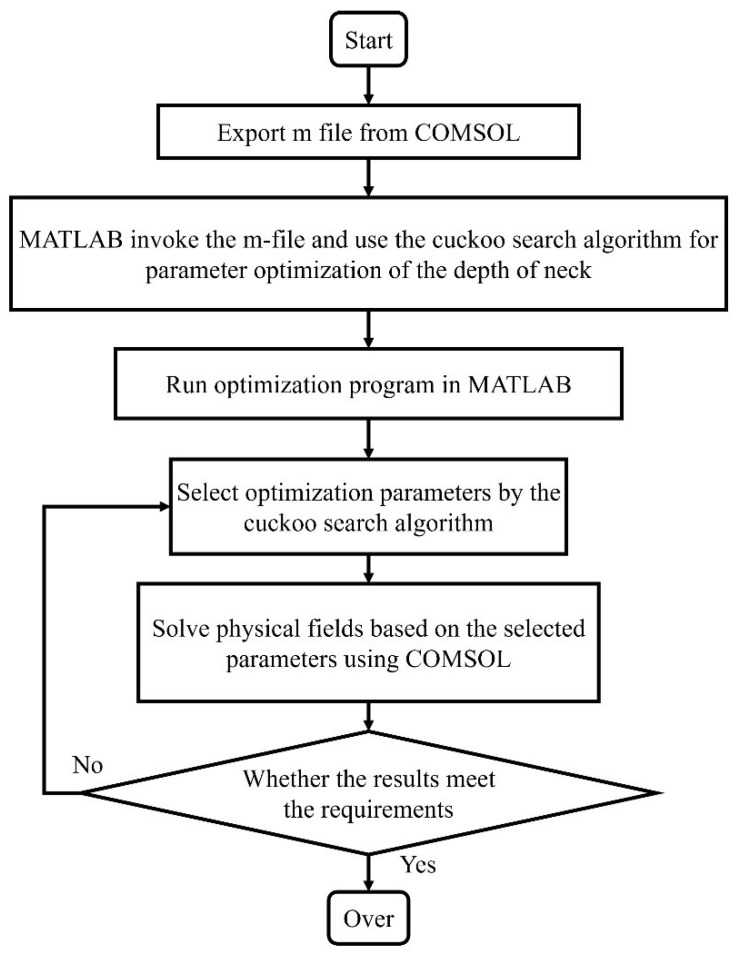
The process of the joint simulation platform for algorithm and physical field simulation.

**Figure 7 materials-15-06450-f007:**
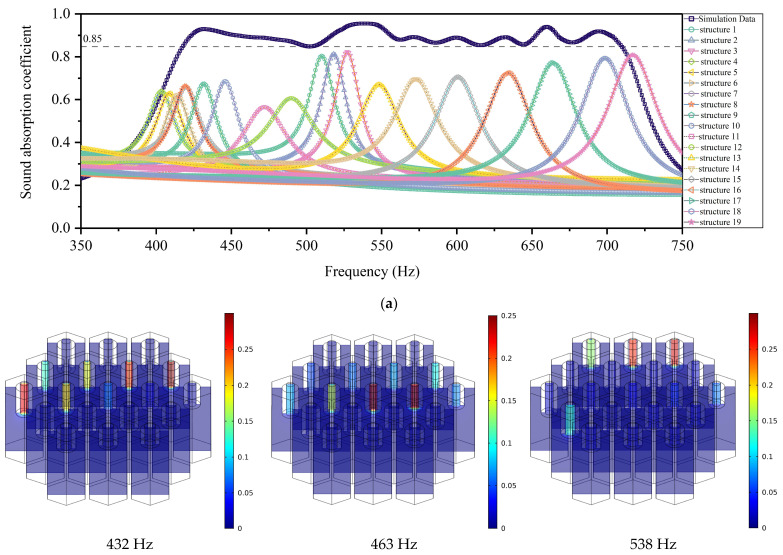
The optimization results. (**a**) The optimized sound absorption coefficients obtained in simulation for the acoustic metamaterial of multiple parallel hexagonal Helmholtz resonators with its corresponding nineteen single structures. (**b**) Instantaneous local velocity of corresponding absorption peaks.

**Figure 8 materials-15-06450-f008:**
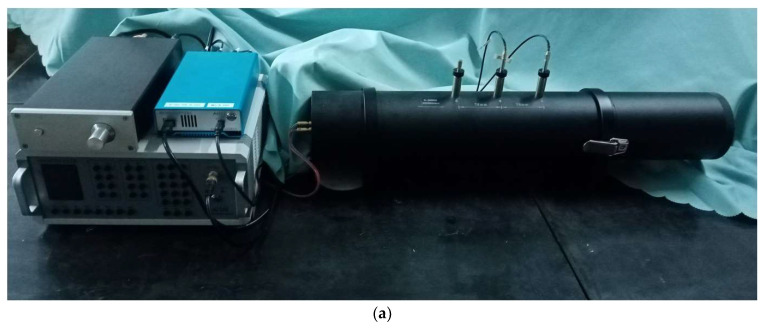
Fabrication and detection of the acoustic metamaterial. (**a**) The AWA6290T transfer function sound absorption coefficient measurement system; (**b**) Schematic diagram of the transfer function tube measurement; (**c**) The low force stereolithography (LFS) 3D printer of Form3; (**d**) The 3D printed sample of acoustic metamaterial with the diameter of 100 mm and the thickness of 40 mm.

**Figure 9 materials-15-06450-f009:**
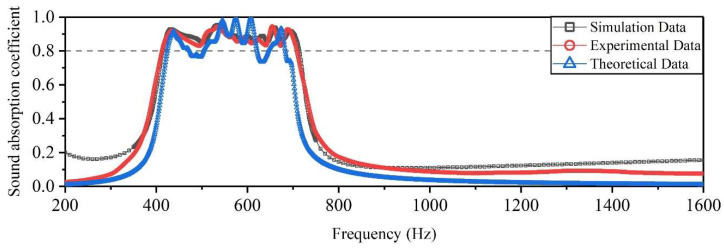
Comparisons of theoretical, simulation and experimental sound absorption coefficients of the proposed acoustic metamaterial of multiple parallel hexagonal Helmholtz resonators.

**Figure 10 materials-15-06450-f010:**
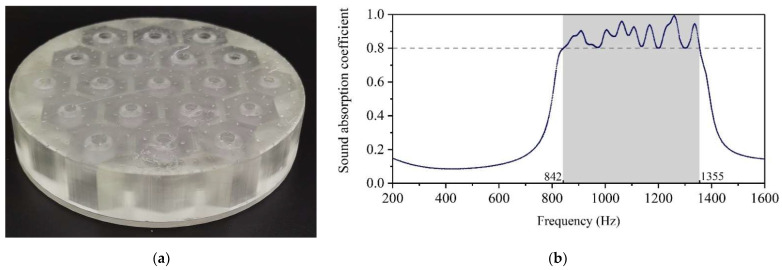
The optimization results of Condition-1. (**a**) Schematic diagram of the 3D printed sample with the diameter of 100 mm and the thickness of 20 mm. (**b**) The corresponding actual sound absorption coefficient.

**Figure 11 materials-15-06450-f011:**
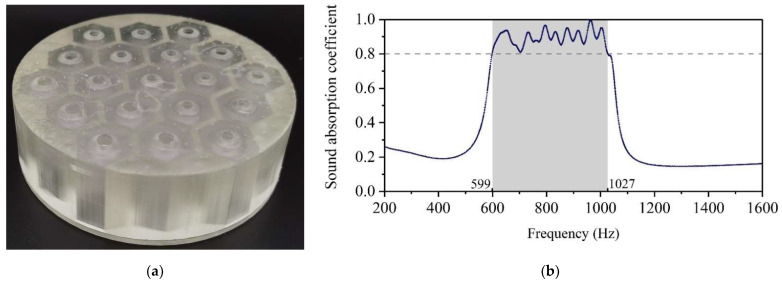
The optimization results of Condition-2. (**a**) Schematic diagram of the 3D printed sample with the diameter of 100 mm and the thickness of 30 mm. (**b**) The corresponding actual sound absorption coefficient.

**Table 1 materials-15-06450-t001:** The initial geometric parameters of the proposed acoustic metamaterial.

Group	Serial Number	Thickness/mm	Diameter of Micropore/mm	Depth of Neck/mm
I	1	40	4.47	13.0
	2			12.1
	3			11.2
II	4		3.54	14.7
	5			13.4
	6			12.3
	7			11.3
III	8		3.75	12.1
	9			11.2
	10			10.3
	11			9.5
	12			8.8
IV	13		4.74	12.3
	14			11.4
	15			10.7
	16			10.0
V	17		5.16	11.8
	18			11.0
	19			10.3

**Table 2 materials-15-06450-t002:** The geometric parameters of the optimized acoustic metamaterial.

Group	Serial Number	Thickness/mm	Diameter of Micropore/mm	Depth of Neck/mm
I	1	40	4.47	15.7
	2			15.2
	3			14.4
II	4		3.54	15.9
	5			15.2
	6			14.7
	7			14.3
III	8		3.75	16.7
	9			15.3
	10			14.1
	11			13.1
	12			11.8
IV	13		4.74	16.2
	14			14.5
	15			12.6
	16			10.9
V	17		5.16	12.0
	18			10.3
	19			9.5

**Table 3 materials-15-06450-t003:** The optimized geometric parameters for design requirements of Condition-1.

Group	Serial Number	Thickness/mm	Diameter of Micropore/mm	Depth of Neck/mm
I	1	20	4.47	8.8
	2			8.2
	3			7.3
II	4		3.54	8.1
	5			7.6
	6			6.8
	7			6.1
III	8		3.75	9.1
	9			8.2
	10			7.1
	11			6.3
	12			5.1
IV	13		4.74	9.0
	14			7.6
	15			6.4
	16			5.3
V	17		5.16	6.2
	18			5.0
	19			4.5

**Table 4 materials-15-06450-t004:** The optimized geometric parameters for design requirements of Condition-2.

Group	Serial Number	Thickness/mm	Diameter of Micropore/mm	Depth of Neck/mm
I	1	30	4.47	10.3
	2			9.1
	3			8.0
II	4		3.54	9.8
	5			9.0
	6			8.4
	7			7.7
III	8		3.75	10.8
	9			9.9
	10			8.6
	11			7.6
	12			6.5
IV	13		4.74	9.5
	14			8.2
	15			7.0
	16			5.9
V	17		5.16	6.4
	18			5.4
	19			4.8

## Data Availability

The data that support the findings of this study are available from the corresponding author upon reasonable request.

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
