# Peer review of "Optimal Design of Acoustic Metamaterial of Multiple Parallel Hexagonal Helmholtz Resonators by Combination of Finite Element Simulation and Cuckoo Search Algorithm"

_materials, 2022, doi:10.3390/ma15186450_

Round 1
Reviewer 1 Report
In the article the description of the experiment is missing. You should complement it with the Chapter ‘Methodology’ and describe what kind of experiment it was, under what conditions it was performed, with what instruments/equipment was it carried out and what is the uncertainty of the experiment.
Also, computer model validation is missing (see the article. Buildings 2022, 12, 347.https: //doi.org/10.3390/buildings12030347)
Literature is well written, but consider supplementing it e.g. with:
Applied Acoustics 2022, 188, 108569, https://doi.org/10.1016/j.apacoust.2021.108569
Journal of Sound and Vibration 2022, 537, 117197,
https://doi.org/10.1016/j.jsv.2022.117197
Author Response
Response to reviewer 1
Thank you very much for your kind review to our manuscript and positive assessment to our research. We have revised the manuscript carefully according to your and other reviewers’ comments. The responses to your comments are as follows.
1. In the article the description of the experiment is missing. You should complement it with the Chapter ‘Methodology’ and describe what kind of experiment it was, under what conditions it was performed, with what instruments/equipment was it carried out and what is the uncertainty of the experiment.
Response:
Thank you very much for your positive assessment and kind suggestion. We have added a chapter ‘Methodology’ in the revised manuscript to describe what kind of experiment it was, under what conditions it was performed, with what instruments/equipment was it carried out and what is the uncertainty of the experiment. Meanwhile, these corresponding modifications were highlighted in yellow in the revised manuscript.
2. Also, computer model validation is missing (see the article. Buildings 2022, 12, 347. https: //doi.org/10.3390/buildings12030347)
Response:
Thank you very much for your positive assessment and kind suggestion. We have modified the manuscript to descript the validation of computer model according to the literature ‘Experimental Validation of the Model of Reverberation Time Prediction in a Room, Buildings 2022, 12, 347’. Meanwhile, this article was added to the list of references in the revised manuscript, which was highlighted in yellow in the revised manuscript.
3. Literature is well written, but consider supplementing it e.g. with:
Applied Acoustics 2022, 188, 108569, https://doi.org/10.1016/j.apacoust.2021.108569
Journal of Sound and Vibration 2022, 537, 117197,
https://doi.org/10.1016/j.jsv.2022.117197
Response:
Thank you very much for your positive assessment and kind suggestion. We have read the two recommended articles ‘Sound transmission loss of a Helmholtz Resonator-based acoustic metasurface, Applied Acoustics 2022, 188, 108569’ and ‘Nonlinear sound absorption of Helmholtz resonators with serrated necks under high-amplitude sound wave excitation, Journal of Sound and Vibration 2022, 537, 117197’ and added in the revised manuscript, which was highlighted in yellow in the revised manuscript.

Reviewer 2 Report
The reviewed article presents the problem of designing resonant metamaterial systems to achieve the desired sound absorbing properties in the assumed noise frequency ranges.
In the reviewer's opinion, the article is interesting and noteworthy, but some elements of the text need to be corrected and clarified.
The style and language of the article needs improvement. For example, in lines 40-41 it is written:
"... the acoustic metamaterial with extra-long physical properties that traditional acoustic materials do not possess has ..." Question arises what are the extra long physical properties?
L. 47-49: "The acoustic metamaterial absorber composed of thick-necked embedded Helmholtz resonators was developed by Zhang [7] achieved perfect sound absorption at 150 Hz."
Shouldn't it be written here: "Developed by Zhang [7] acoustic metamaterial absorber composed of thick-necked embedded Helmholtz resonators achieved perfect sound absorption 48 at 150 Hz."?
L. 63-64: "... adopted the particle swarm optimization to optimize the depth..." ...
L. 75-77: "Due to the immature research on the acoustic absorption mechanism of acoustic metamaterial, a number of simplifications are 76 usually taken in the process of theoretical modelling, which resulted in large errors between the theoretical results and experimental results." This sentence should be followed by a further literature review on this subject.
The structure of the article is unclear. The section preceding the results of the study introduces information that appears to be the results of the study (Lparagraph 3.4 that is appearing before paragraph 4).
The reviewer suggests implementing the full code which is generally discussed in paragraph 3.3 in the appendix of the article.
Descriptions of similar research to the work done by the authors can be found in the literature. In the reviewer's opinion, it is important to clearly indicate the new addition to the field of research and the limitations of this work.
Yours Sincerely,
Reviewer.
Author Response
Response to reviewer 2
Thank you very much for your kind review to our manuscript and helpful assessment to our research. We have revised the manuscript carefully according to your and other reviewers’ comments. The responses to your comments are as follows.
1. The reviewed article presents the problem of designing resonant metamaterial systems to achieve the desired sound absorbing properties in the assumed noise frequency ranges.
In the reviewer's opinion, the article is interesting and noteworthy, but some elements of the text need to be corrected and clarified.
Response:
Thank you very much for your positive assessment and kind suggestion. We have corrected the whole manuscript according to your and other reviewers’ comment, which aim to make the manuscript more reasonable and readable.
2. The style and language of the article needs improvement. For example, in lines 40-41 it is written:
"... the acoustic metamaterial with extra-long physical properties that traditional acoustic materials do not possess has ..." Question arises what are the extra long physical properties?
47-49: "The acoustic metamaterial absorber composed of thick-necked embedded Helmholtz resonators was developed by Zhang [7] achieved perfect sound absorption at 150 Hz."
Shouldn't it be written here: "Developed by Zhang [7] acoustic metamaterial absorber composed of thick-necked embedded Helmholtz resonators achieved perfect sound absorption 48 at 150 Hz."?
63-64: "... adopted the particle swarm optimization to optimize the depth..." ...
75-77: "Due to the immature research on the acoustic absorption mechanism of acoustic metamaterial, a number of simplifications are 76 usually taken in the process of theoretical modelling, which resulted in large errors between the theoretical results and experimental results." This sentence should be followed by a further literature review on this subject.
Response:
Thank you very much for your kind comment and significant suggestion. We have corrected the corresponding presentations in the revised manuscript according to the mistakes you have pointed out. Meanwhile, the whole manuscript was improved carefully to eliminate the errors in style and language, and these revisions were highlighted in yellow in the revised manuscript as well.
3. The structure of the article is unclear. The section preceding the results of the study introduces information that appears to be the results of the study (Lparagraph 3.4 that is appearing before paragraph 4).
Response:
Thank you very much for your kind comment and significant suggestion. The structure of the manuscript was adjusted according to your and the other reviewers’ comment to make this manuscript more reasonable, and these adjustments were highlighted in yellow as well.
4. The reviewer suggests implementing the full code which is generally discussed in paragraph 3.3 in the appendix of the article.
Response:
Thank you very much for your kind suggestion. The section ‘3.3. Interactive operations with MATLAB and COMSOL’ described the joint simulation platform for algorithm and physical field simulation, which was the major innovation of this manuscript. This method had been applied a patent for invention and it is under substantive review right now, so it was inconvenient to supply the full code of this method in the main text or appendix of this manuscript.
5. Descriptions of similar research to the work done by the authors can be found in the literature. In the reviewer's opinion, it is important to clearly indicate the new addition to the field of research and the limitations of this work.
Response:
Thank you very much for your kind suggestion. It is well known that the Helmholtz resonators with extended aperture depth have been a favorable choice for low frequency noise absorption, which make them the research highlights in the field of sound absorption. The major innovation of this manuscript was the joint simulation platform of MATLAB and COMSOL for the algorithm and physical field simulation, which aimed to improve the optimization accuracy and efficiency. According to your comment, the new addition of this manuscript to the field of research and the limitations of this work were added in the revised manuscript and these modifications were highlighted in yellow, which aimed to make the manuscript more persuasive.

Reviewer 3 Report
The manuscript addresses the optimization of an acoustic metamaterial, composed of multiple parallel hexagonal Helmholtz resonators with sub-wavelength dimensions, for maximizing its sound absorption.
The structure of the manuscript is adequate, and both the methodology and results are well presented. The statements and conclusions in the text are supported by the results, and the amount and relevance of the provided references is also adequate.
Some aspects should be addressed prior to the final acceptance and publication of the manuscript:
1. The state of the art can also include some other relevant references, such as:
a. Romero-García, V., Theocharis, G., Richoux, O., Merkel, A., Tournat, V. and Pagneux, V. (2016). Perfect and broadband acoustic absorption by critically coupled sub-wavelength resonators. Scientific Reports 6: 19519. https://doi.org/10.1038%2Fsrep19519.
b. Herrero-Durá, I., Cebrecos, A., Picó, R., Romero-García, V., García-Raffi, L.M. and Sánchez-Morcillo, V.J. (2020). Sound absorption and diffusion by 2D arrays of Helmholtz resonators. Applied Sciences, 10(5), 1609. https://doi.org/10.3390/app10051690.
2. Line 104: Figure 3 should be referenced in this point, as the authors introduce the sound absorption coefficient of the initial structure.
3. Line 132: The authors indicate that the perforation was ensured to be different between groups in a range of 0.5-0.9%. How was this checked carried out? In a “visual” way, imposing different starting points of the optimization process for each group, etc.
4. Line 136: The wall thickness was taken to be 2 mm. This thickness is indeed very small and could lead to vibroacoustic effects in the system. Were these phenomena observed? The authors could include the walls of the resonators in the numerical model and do some initial checking (not to include in the manuscript).
5. Section 3.1: Could the authors provide some more information about the numerical model (e.g., meshing criteria, etc.)? What was the size of the considered PML?
Author Response
Response to reviewer 3
Thank you very much for your kind review to our manuscript and positive assessment to our research. We have revised the manuscript carefully according to your and other reviewers’ comments. The responses to your comments are as follows.
1. The manuscript addresses the optimization of an acoustic metamaterial, composed of multiple parallel hexagonal Helmholtz resonators with sub-wavelength dimensions, for maximizing its sound absorption.
The structure of the manuscript is adequate, and both the methodology and results are well presented. The statements and conclusions in the text are supported by the results, and the amount and relevance of the provided references is also adequate. Some aspects should be addressed prior to the final acceptance and publication of the manuscript.
Response:
Thank you very much for your positive assessment and kind suggestion. We have corrected the whole manuscript according to your and other reviewers’ comment, which aim to make the manuscript more reasonable and readable.
2. The state of the art can also include some other relevant references, such as:
a. Romero-García, V., Theocharis, G., Richoux, O., Merkel, A., Tournat, V. and Pagneux, V. (2016). Perfect and broadband acoustic absorption by critically coupled sub-wavelength resonators. Scientific Reports 6: 19519.
b. Herrero-Durá, I., Cebrecos, A., Picó, R., Romero-García, V., García-Raffi, L.M. and Sánchez-Morcillo, V.J. (2020). Sound absorption and diffusion by 2D arrays of Helmholtz resonators. Applied Sciences, 10(5), 1609. https://doi.org/10.3390/app10051690.
Response:
Thank you very much for your kind suggestion. We have read the two recommended two articles ‘Perfect and broadband acoustic absorption by critically coupled sub-wavelength resonators. Scientific Reports 2016, 6: 19519.’ and ‘Sound absorption and diffusion by 2D arrays of Helmholtz resonators. Applied Sciences 2020, 10(5), 1609’ and added in the revised manuscript, which was highlighted in yellow in the revised manuscript.
3. Line 104: Figure 3 should be referenced in this point, as the authors introduce the sound absorption coefficient of the initial structure.
Response:
Thank you very much for your kind suggestion. For the Line 104 in the original manuscript, the given frequency range of 487-675 Hz was the test results of major noise frequencies in a workshop under the operation, and it was not the sound absorption coefficients of the initial structure for the acoustic metamaterial. Moreover, the Figure 3 showed the sound absorption coefficient of the initial structure gained by finite element simulation and the gray area represents the frequency range that met the design requirements.
4. Line 132: The authors indicate that the perforation was ensured to be different between groups in a range of 0.5-0.9%. How was this checked carried out? In a “visual” way, imposing different starting points of the optimization process for each group, etc.
Response:
Thank you very much for your kind comment. Influence of the perforation rate to the sound absorption performance of the multiple parallel connection Helmholtz resonators had been investigated with a double resonators by the finite element simulation in our another article entitled ‘Development of Adjustable Parallel Helmholtz Acoustic Metamaterial for Broad Low–frequency Sound Absorption Band’, which has been accepted by Materials and waiting for publication right now. The 3–dimensional finite element simulation model for the double resonators in this article was shown in the Figure 1.
(The figure cannot be inserted here, so please see the figure in the attached PDF file)
Figure 1. 3–dimensional finite element simulation model for the double resonators. (a) 3D structure of the whole model; (b) The vertical view of model with the relevant parameters; (c) The front view of model with the relevant parameters; (d) The finite element simulation model with the gridded mesh.
Influence of perforation ratio σ to sound absorption performance of the double resonators was investigated by changing diameter of the front panel D, and the studied perforation ratio σ was 0.3%, 0.5%, 0.7%, 0.9% and 1.1% respectively, as shown in the Figure 2. It could be observed that the peak sound absorption coefficient αmax was 0.8052, 0.9538, 0.9972, 0.9942 and 0.9695 along with increase of perforation ratio σ from 0.3% to 1.1%, and each resonator frequency f0 was around the 604 Hz. The perforation ratio had little influence to resonator frequency f0 and it significantly affected the peak sound absorption coefficient αmax. The best peak sound absorption coefficient αmax was obtained when the perforation ratio σ was around 0.7%, which could gain the perfect absorption with sound absorption coefficient close to 1.
(The figure cannot be inserted here, so please see the figure in the attached PDF file)
Figure 2. Influence of perforation ratio to sound absorption performance of double resonators.
Influence of diameter of the aperture was investigated and the diameter of the aperture 2 d2 was selected as the studied variable. Its value was selected as 1.7 mm, 2.2 mm, 2.7 mm, 3.2 mm and 3.7 mm respectively. Except diameter of the aperture 2 d2, the other parameters were selected as d1=2.7 mm, l1=l2=6 mm, L1=L2=50 mm, a1=a2=10 mm, and σ2=0.7%. Thus, the corresponding perforation ratio for resonator 1 σ1 was 1.76%, 1.05%, 0.70%, 0.50% and 0.37% respectively. Influence of diameter of the aperture to sound absorption performance of the double resonators in simulation was shown in the Figure 4. It could be found that except the condition of d1=d2=2.7 mm, all the other 4 conditions could obtain the double sound absorption peaks.
(The figure cannot be inserted here, so please see the figure in the attached PDF file)
Figure 3. Influence of diameter of the aperture to sound absorption performance of the double resonators in simulation.
In order to avoid the duplication in the presentation, these contents were not presented in this manuscript, and the article ‘Yang, X.; Yang, F.; Shen, X.; Wang, E.; Zhang, X.; Shen, C.; Peng, W. Development of Adjustable Parallel Helmholtz Acoustic Metamaterial for Broad Low-Frequency Sound Absorption Band. Materials 2022, 15, 5938.’ was added in the list of references, which could be treated as certification to the selection of perforation rate for multiple parallel connection Helmholtz resonators in this manuscript.
5. Line 136: The wall thickness was taken to be 2 mm. This thickness is indeed very small and could lead to vibroacoustic effects in the system. Were these phenomena observed? The authors could include the walls of the resonators in the numerical model and do some initial checking (not to include in the manuscript).
Response:
Thank you very much for your kind suggestion. In our former research, the vibroacoustic effect had not been taken into consideration and the wall was treated as the rigid wall in the finite element simulation process. In the experimental process, the sample was set in the tube and it was impossible to observe whether the wall had some vibration or not. Your suggestion is very meaningful, and we will check thickness of the wall to sound absorption performance of the acoustic metamaterial by finite element simulation in the future study.
6. Section 3.1: Could the authors provide some more information about the numerical model (e.g., meshing criteria, etc.)? What was the size of the considered PML?
Response:
Thank you very much for your kind comment. Some more parameter settings in the finite element simulation process were added in the revised manuscript, which aimed to make the manuscript easy to understand. The thickness of the perfect match layer (PML) was 1.5 times of that of background pressure field (BPF), and their sizes were determined by total thickness of the acoustic absorber and the analyzed frequency range simultaneously. Normally speaking, their sizes would be larger when the total thickness of the acoustic absorber is larger or the analyzed frequency range was in a lower region.

Round 2
Reviewer 1 Report
The authors dealt well with all my comments.
Author Response
Response to reviewer 1
Thank you very much for your kind review to our manuscript and positive assessment to our research. We have revised the manuscript carefully according to your and other reviewers’ comments. The responses to your comments are as follows.
1. The authors dealt well with all my comments.
Response:
Thank you very much for your positive assessment. We have further corrected the whole manuscript about the English spelling and grammar, which aims to make the manuscript more reasonable and readable.

Reviewer 2 Report
In my opinion, the article can be published after a minor revisions, as noted below:
1. the language and style of the text should be refined,
2. the components of the measurement system shown in Figure 8 should be described clearly - fig. 8a shows several devices, what is each device responsible for? it is worth introducing a description in the picture; fig. 8b shows rather the housing of the device and adds little to the description of the methodology.
Best regards,
Reviewer.
Author Response
Response to reviewer 2
Thank you very much for your kind review to our manuscript and helpful assessment to our research. We have revised the manuscript carefully according to your and other reviewers’ comments. The responses to your comments are as follows.
1. In my opinion, the article can be published after a minor revisions. The language and style of the text should be refined.
Response:
Thank you very much for your positive assessment and kind suggestion. We have further corrected the whole manuscript about the English spelling and grammar, which aims to make the manuscript more reasonable and readable.
2. The components of the measurement system shown in Figure 8 should be described clearly - fig. 8a shows several devices, what is each device responsible for? It is worth introducing a description in the picture; fig. 8b shows rather the housing of the device and adds little to the description of the methodology.
Response:
Thank you very much for your kind comment and significant suggestion. We have added a new figure about schematic diagram of the transfer function tube measurement, as shown in the Figure 8b in the revised manuscript. Meanwhile, the operating principle of the low force stereolithography (LFS) 3D printer of Form3 in the Figure 8c (corresponded to the Figure 8b in the original manuscript) was descripted.
The main text of the section “4.3.1 Methodology” is as follows:
To verify the sound absorption effect of the acoustic metamaterial, experiments were conducted with the AWA6290T transfer function sound absorption coefficient measurement system (supported by Hangzhou Aihua Instruments Co., Ltd., Hangzhou, China), as shown in the Figure 8a, which could detect the sound absorption coefficients of sound absorbing material or structures with normal incidence according to GB/T 18696.2-2002 (ISO 10534-2:1998) “Acoustics–Determination of sound absorption coefficient and impedance in impedance tubes–part 2: Transfer function method”, and its schematic diagram was shown in the Figure 8b [36–41]. A cylindrical sample with a diameter of 100 mm was manufactured by the low force stereolithography (LFS) 3D printer of Form3 (supported by the Formlabs Inc., Summerville, MA, USA), as shown in the Figure 8c, and the prepared sample for the investigated acoustic metamaterial was exhibited in the Figure 8d. The proposed acoustic metamaterial of the multiple parallel hexagonal Helmholtz resonators was modeled in the 3D modeling software, and it was further introduced into the Preform software supported by Form3 3D printer. When fabrication of the sample was finished, it was further cleaned by the Formlabsform wash (Formlabs Inc., Boston, MA, USA) to remove residual liquid resin and irradiated for solidification by the Formlabs-Form Cure (Formlabs Inc., Boston, MA, USA). The used photosensitive liquid resin in this research was ClearV4, which was purchased from the self–support flagship store of Formlabs 3D printer in JD.com (JD.com Inc., Beijing, China). The acoustic metamaterial made by photosensitive resin 3D printing had a smooth surface and well hardness, which met experimental requirements [36–41]. The AWA6290T detector consisted of the AWA5871 power amplifier, the AWA6290B dynamic signal analyzer, the AWA8551 impedance tube, and corresponding analysis software in the workstation, as shown in the Figure 8b. The analysis software could finish the 1/3 OCT analysis and fast Fourier trans-form (FFT) analysis. Meanwhile, the original incident acoustic wave was also controlled by the signal generation software in the workstation. The detected sample was fixed in the end of the impedance tube and two sensors were utilized to detect the signal of the incident and reflected acoustic waves. The distance between sensor 1 and sensor 2 was selected as 70 mm. The detected frequency range was 200–1600 Hz and there was 1502 sampling frequency points in this range. Moreover, for the purpose of elimination of the accidental error, the detection was repeated for 200 times for each sampling frequency point, and the final data was average of the 200 values obtained in the 200 times of measurement. The measurement process was full-automatic, which only took no more than 1 minute for once detection program [20].
